# CHOPS: CHat with custOmer Profile Systems for Customer Service with LLMs

**Jingzhe Shi**[1,4]**, Jialuo Li**[1,4]**, Qinwei Ma**[1,4]**, Zaiwen Yang**[1,4]**, Huan Ma**[1,4]**, Lei Li**[✉][2,3,4]

[1] Tsinghua University, [2] University of Copenhagen, [3] University of Washington, [4] CPHOS*

[1] {shi-jz21, lijialuo21, mqw21, yangzw23, mah21}@mails.tsinghua.edu.cn

[2] lilei@di.ku.dk

## Abstract

Businesses and software platforms are increasingly using large language models (LLMs) such as GPT-3.5, GPT-4, GLM-3, and LLaMa-2 as chat assistants with file access or as reasoning agents for custom service. Current LLM-based customer service models exhibit limited integration with customer profiles and lack operational capabilities, while existing API integrations prioritize diversity over precision and error avoidance, which are crucial in real-world scenarios for Customer Service. We propose an LLMs agent called **CHOPS** (**CH**at with cust**O**mer **P**rofile in existing **S**ystem) that: (1) efficiently utilizes existing databases or systems to access user information or interact with these systems based on existing guidance; (2) provides accurate and reasonable responses or executing required operations in the system while avoiding harmful operations; and (3) leverages the combination of small and large LLMs together to provide satisfying performance while having decent inference cost. We introduce a practical dataset, *CPHOS-dataset*, including a database, some guiding files, and QA pairs collected from *CPHOS**. We conduct extensive experiments to validate the performance of our proposed **CHOPS** architecture using the *CPHOS-dataset*, aiming to demonstrate how LLMs can enhance or serve as alternatives to human customer service. Code for our proposed architecture and dataset can be found at https://github.com/JingzheShi/CHOPS.

## 1 Introduction

In most organizations with human customer service, there is usually a system storing customer information. Responses are based on the customer's profile, like user type and purchase history, following set guidelines. Customer service can also update the customer's status upon request. Large language models (LLMs), such as GPT-3.5 (OpenAI, 2022), GPT-4.0 (OpenAI, 2023), GLM (Zeng et al., 2022; Du et al., 2022) and LLaMa (Touvron et al., 2023), have emerged as a representative achievement of AI development in the past decade. With their mastering of common knowledge and their ability to understand prompts and generate contextually relevant answers, LLMs have been used as assistants across a wide range of application scenarios, including chatting assistants, coding assistants, automatic assistant agents, etc (Kalla & Smith, 2023).

A common paradigm for equipping LMs with external knowledge while avoiding long context lengths that are costly or technically hard is RAG (Lewis et al., 2021), represented by the widely used and efficient Vector Database using a sentence embedding model such as the Universal Sentence Encoder (Cer et al., 2018). Most publicly available architectures for utilizing LLMs as customer service mainly follow this paradigm, e.g. Databricks (Databricks) enables users to upload guiding files thus building a Vector Database-based customer service agent to augment human customer service. However, for most software platforms

---

*CPHOS is an **academic non-profit organization** that employs an online platform to facilitate the organization of simulated Physics Olympiads for high school teachers and students. More information at: https://cphos.cn.

or businesses, to answer based on a series of guides is not enough: to query information or to manipulate the system using a set of APIs is necessary in some scenarios.

Figure 1: Left: Existing scenarios for Customer Service require File QA and System Manipulation. Middle: Possible mistakes in Customer Service. Accuracy is needed in this scenario, especially to avoid those harmful operations. Right: existing methods to use LLM as assistants. LLMs for APIs like ToolLLM (Qin et al., 2023) mainly focus on a large number of APIs in API hubs.

Previous works (involving models, agent architectures, and datasets) on LLMs using APIs (Patil et al., 2023; Qin et al., 2023; Tang et al., 2023) mainly focus on the LLM's ability to choose between a vast amount (100 through 100,000+) APIs and to accomplish different tasks. However, for a particular customer service scenario, much fewer APIs are needed but high accuracy is needed, especially for modifying user status in important aspects like banking. This difference in the focus point of Customer Service and previously defined API-using tasks sheds light on a different method needed for Customer Service compared to previous API-using methods and new datasets for evaluation of such methods.

Online Customer Service, with its long history since the popularization of the Internet and personal computers, requires accurate answers or modifications to user profiles in customer service based on the user's questions or requests and the guiding file. Unlike other scenarios for reasoning or prompting, especially on math problem solving (Wei et al., 2023; Yao et al., 2023; Zhang et al., 2023), or utilizing numbers of APIs (Patil et al., 2023; Qin et al., 2023; Tang et al., 2023), we focus on the new task of customer service, which sheds light on the accuracy and admittance of the answer; moreover, the cost of the LLM needs to be controlled.

Previous efforts to integrate Language Models (LMs) with databases have focused on generating SQL commands, which poses risks such as incorrect or harmful commands due to LMs' hallucination issues (Huang et al., 2023). To address this, we employ APIs for database management, steering clear of direct SQL command generation. Platforms like LangChain (Chase, 2022), Gorilla (Patil et al., 2023), and ToolLLM (Qin et al., 2023) offer extensive API libraries, which may exceed the needs of specific customer service applications where the accuracy and proper use of a limited set of APIs are paramount. Unlike scenarios requiring thousands of APIs, customer service systems prioritize precise and correct API usage, highlighting the importance of quality over quantity in API interactions.

Leveraging insights, shown in Figure. 3, from studies on using LLMs to verify responses of other LLMs, we introduce an Executor-Verifier architecture, employing a verifier agent to assess and ensure the validity of commands executed by another LLM, termed the executor agent as Figure 3. This approach aims to enhance the accuracy of integrating LLMs with APIs by reiterating the execution process based on the verifier's feedback.

To address challenges when user requests involve both API use and information from guiding files, we evolved our approach into a Classifier-Executor-Verifier architecture, enhancing efficiency and reducing inference costs. This architecture first classifies user requests to determine if they require access to APIs, guiding files, or both, thus avoiding unnecessary processing and long, redundant texts from guiding files for API-only queries. Additionally, by segmenting guiding files into smaller, more focused documents and employing a more nuanced classifier, we further minimize inference costs by ensuring that only the most relevant sections are utilized. This tailored architecture, specifically designed for Customer Service scenarios involving guiding files and user systems, optimizes performance while conserving computational resources.

In response to the lack of datasets for customer service that incorporate details on internal guiding files or systems for API interactions within the customer service domain, we propose the *CPHOS-dataset*. This dataset, derived from real-world scenarios at the *Cyber Physics Olympiad Simulations (CPHOS)*, a non-profit organization, includes databases, guiding files in PDF format, and QA pairs from actual interactions, aiming to bridge the gap in existing resources for LLM research in customer service. Carefully curated and anonymized, the *CPHOS-dataset* serves as a comprehensive tool for evaluating the effectiveness of LLMs in customer service environments, especially those involving complex systems and guiding documents (Vector, 2018; McAuley; Rajpurkar et al., 2016; Joshi et al., 2017; Patil et al., 2023; Qin et al., 2023; Tang et al., 2023; OpenAI, 2022; 2023).

To this end, we propose a general framework called **CHOPS**: **CH**at with cust**O**mer **P**rofile in existing **S**ystem as Figure 3 shown. In our **CHOPS** framework, we propose a classifier-executor-verifier based framework that makes the LLM better utilize tools, especially Database and guidance. Through the work of these three agents, the task of answering a user's question is decomposed into (1) classifying the type or theme of the user's question, (2) giving answers or decision operations to be executed and (3) verifying then reject or commit the result of the executor.

Our work makes several contributions to the application of LLMs, especially in the field of customer service with LLMs.

- We propose a framework **CHOPS** to embed LLM safely, effectively, high-cost-performancely into existing customer service systems.
- Our experiments demonstrate that by using weaker LLMs in our architecture, we can achieve significantly better performance compared to naively using stronger LLMs while saving cost. Moreover, using gpt-4 as Executor while gpt-3.5-turbo as Classifier and Verifier can reach 98% accuracy with decent cost.
- We proposed *CPHOS-dataset*, it is a practical dataset collected in real scenarios that can be used to validate methods utilizing LLMs as customer service.

## 2   Related Works

**Retrieval-Augmented Generation with LLMs.**   Incorporating external knowledge sources into LLMs for enhanced performance on knowledge-intensive tasks has seen advancements through Retrieval-Augmented Generation (RAG), with the use of vector-based databases for PDF files being a notable example (Tripathi, 2023). This approach encodes user queries into vectors, using k-nearest neighbors (KNN) to retrieve relevant information. In customer service, LLMs augmented with large databases have aimed to provide encyclopedic support for user inquiries (dat, 2022; Wulf, 2022; cms, 2022). However, such methods sometimes struggle in scenarios requiring modifications to a user's profile within existing systems, a crucial aspect of customer service. This highlights the importance of integrating LLMs with software systems for direct interaction tasks, essential for operational efficiency in businesses.

**LLM Agents.**   The research on LLMs as specialized agents is an evolving field in artificial intelligence and natural language processing. Initial research focused on using predefined prompts or fine-tuning to enhance LLMs for specific tasks, establishing their potential in

specialized applications like natural language understanding. Recent studies (Hu et al., 2023a; Doe & Smith, 2023) explore LLMs in agent-based architectures for complex problem solving, such as mathematical puzzles, highlighting the importance of architectural design in improving performance. This progression suggests new avenues for leveraging LLMs in agent-based systems, offering insight into their capabilities for advanced interaction and problem-solving tasks.

**LLMs tools.** Recent research has explored the enhancement of LLMs with external tools to improve their performance for a variety of tasks. Studies like OpenAI (2023); Hu et al. (2023b) demonstrate that equipping LLMs with tools, including automated programming interfaces, database management systems, and coding environments, can significantly expand their capabilities. These advances illustrate the potential for LLMs to generate more accurate code, perform advanced database queries, and overall broaden their applicability by leveraging specialized tools.

# 3 *CPHOS-dataset:* A real-scene dataset for customer service

The *CPHOS-dataset* is collected from an online platform of *CPHOS*, a non-profit organization dedicated to holding Simulated Physics Olympiads online through the online platform as Figure 2 shown.

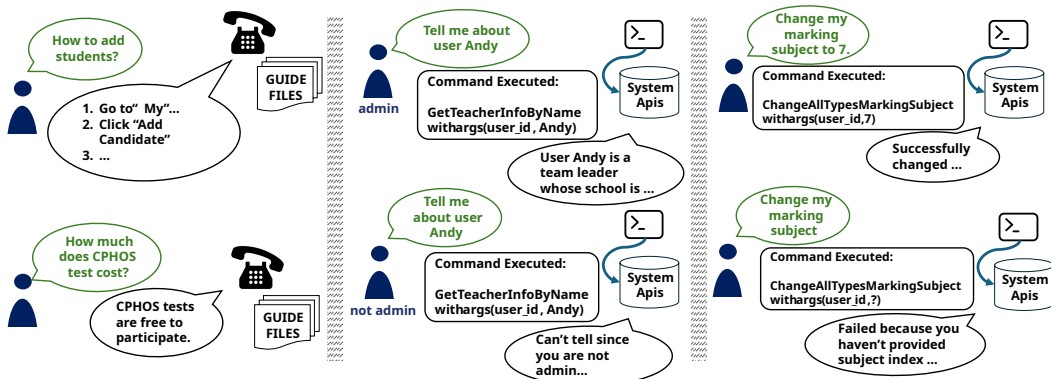

Figure 2: Dataset Examples include guide file-related QAs on the left; in the middle and right, there are system-related QAs and instructions. For the same query, results may differ based on the Query User Status (middle). Similarly, for the same API, the outcome of calling it may vary.

## 3.1 Database

The online system of the Simulated Olympiad utilizes a MySQL database. We provided 9 data desensitized tables. The detailed description can be found in the appendix A.1.

In short, given a user's nickname, one can do a series of queries on tables in the database to obtain or modify partially the profile of the user. The most important field of a user profile includes (0) approved_to_use_online_platform; (1) user_name; (2) school_id; (3) user type: team leader, vise team leader, arbiter; (4) marking_question_id, etc.

Unlike previous works (Hu et al., 2023a) directly using LLMs to generate SQL commands for query or modification to the database, **we wrapped the query and modification into a series of python APIs**, following the idea of Repository Pattern in software design. We provide 9 Data Managing APIs and 18 Data Query APIs, 10 of which are available to LLMs. We collect instructions and queries to the system and augment them with GPT-4 into 104 System-related queries and instructions. There are several advantages of wrapping SQL commands and LLMs manipulate databases through these APIs, in that:

- Properly named APIs are much easier for LLMs to understand and to generate compared to complicated table structures and SQL commands for the database.

- By limiting APIs LLMs can use, or by checking status inside the APIs, one can prevent unwanted or harmful operations that might be carried out by LLMs in extreme conditions.

- In codes of software or websites that are written following the Repository Pattern (or more broadly, the principle of 'encapsulation' in software architecture), a series of pre-defined APIs for a database is likely to exist. Much less effort is needed to modify these APIs into APIs suitable for LLMs than to check for correct SQL commands generated by LLMs.

The diversity of the APIs not only comes from the number of APIs. Calling them from a different user and with different arguments would give different results as shown in the middle and right part of Figure 2.

## 3.2 PDF-based guides

There are two main guiding files provided by *CPHOS*: the mini-program guiding file and questions that are commonly asked by users, together with their answers. These QA files, together with the mini-program guiding file, are what we refer to as PDF-based guides. All files are translated by us into English. In practice, we merge them into one file for RAG. Full PDF-based guides are appended in the supplementary materials. Collected QA pairs from real scene, we augment them through GPT-4 and repetition into 102 QA pairs on Guide Files, an example of which can be shown in the left part of Figure 2.

## 4 Methods

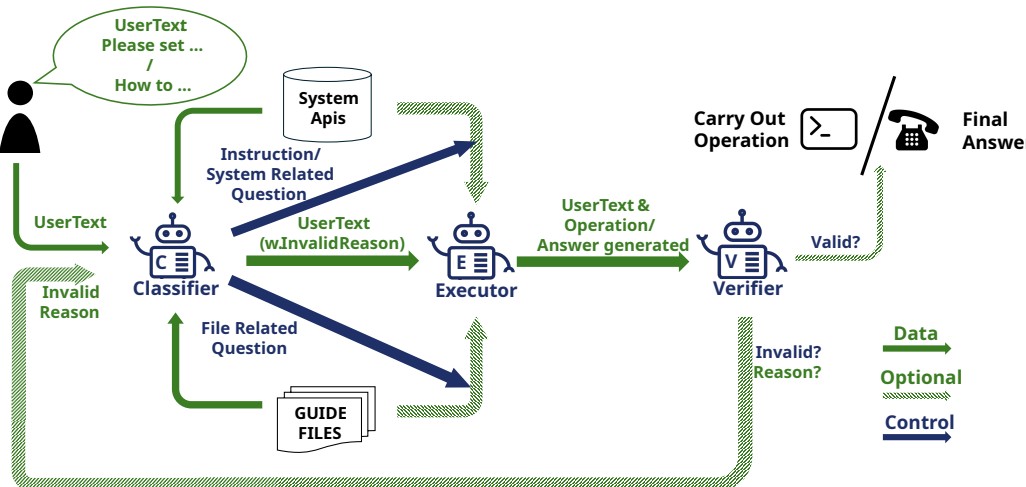

Figure 3: Our **CHOPS** architecture including Classifer, Executor and Verifier.

Given the previous setup, in Figure 3, we define our task as follows. Given a user's nickname and its question, the task is to give a proper answer or an appropriate execution command to the system, based on the status of the user in the existing system and the guiding files.

## 4.1 Framework Overview

We propose a three-agent architecture for this task, termed the classifier-executor-verifier architecture. In this architecture, each agent is implemented using a Large Language Model (LLM). This three-stage architecture functions as follows:

C: The Classifier is given the UserTexts, the System API descriptions and several relevant (and short) chunks from the guiding files. **The Classifier classifies the UserTexts based on information needed for the following pipeline**. The classifier itself does not output all relevant information but only indicates the type of information that would be beneficial for further processing.

E: The Executor is given the UserTexts and other information that the classifier classifies as helpful in the Classifier Stage. Note that the information can be richer than that given to the Classifier (e.g. longer retrieved chunks, more detailed descriptions, etc.). **The Executor then gives a proposed answer or API call based on the information given**.

V: **The Verifier is responsible for revisiting the Executor's result, verifying if it is valid or not and giving a reason as well.** If valid, then the answer is returned or the execution is carried out, and a reply is generated based on the result of that execution. If invalid, then the whole process will be redone, while the Classifier and Executor can see the invalid reason provided by the verifier as a reference.

### 4.1.1 Input Classifier

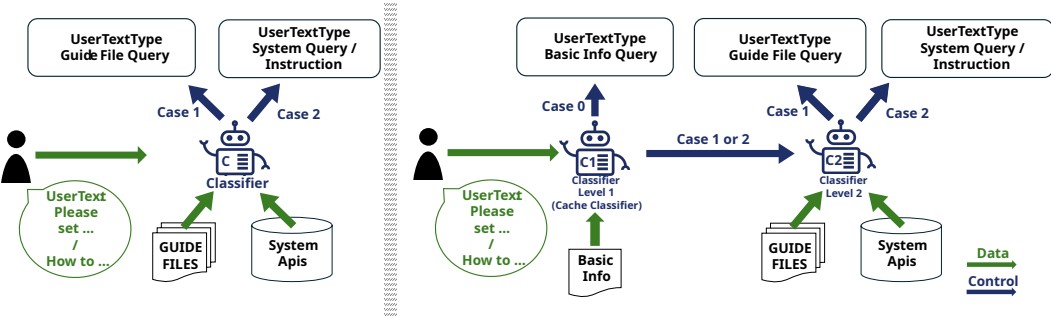

Figure 4: Classifier Architecture. Left: a binary 1-level Classifier. Right: a 2-level Classifier

Previous work (Chen et al., 2023) has shown that the longer and more complicated the retrieved content, the more difficult it is for the Executor LLM to find the exact piece of information and return it. Also, feeding the Executor every time with retrieved chunks from guide file and API info is token-consuming. We utilize a Classifier to classify the information domain that needs to be given to the Executor in advance to solve these two issues.

A simple and experimentally-proved effective and efficient design for the Classifier is a binary classifier, or a 1-level Classifier as shown in the left part of Figure 4. This classifier only chooses between two categories: (1) that the UserTexts are about a query to the guiding file, and (2) that the UserTexts are related to a query or an instruction to the system. Non-classifiable cases, which are not related to the above two cases, are dealt with in the same way as (1) in the following pipeline. Note that although we use the same RAG method to retrieve file chunks for the Classifier and the Executor, the chunk length for Classifier is set to be lower to save inference cost.

Moreover, we observe that many queries and questions are about basic information of the user in the system that is much shorter and less token-consuming compared to the retrieved chunks. Inspired by the idea of **Cache**, we further add one categorized: (0) 'Basic Info' apart from (1) 'Guide File', (2) 'System API'. This leads to the development of a two-level classifier architecture shown in the right part of Figure 4. One classifier will first decide whether the UserTexts are solvable only given the Basic Information (without the need for 'Guide Files' and 'System APIs'). If it asserts yes, then the pipeline will go on and the Executor will only see the Basic Information. If not, then it goes to the second level classifier to categorize UserTexts into class (1) or (2), and further provide the corresponding information to the Executor. The experiment indicates this architecture improves accuracy while saving token consumption.

### 4.1.2 Executor

The Executor needs to return an answer or give an appropriate execution command given information provided. In practice, we do prompt-tuning separately for different cases given by the Classifier.

### 4.1.3 Verifier

Previous works (Weng et al., 2023; Zhang et al., 2023) have proved the effectiveness of a Verifier for re-checking correctness.

In our work, the Verifier verifies the result and, if valid, summarizes the answer or generates a response to the user based on the executed operation.

Like the Executor, we do prompt tuning separately for different cases given by the Classifier. For the Guide File cases, we feed the retrieved results to the Verifier as well.

The Verifier is required to output a valid score (1-10) and reasons at the same time. If the verification result is invalid, then we redo the whole C-E-V process while giving the Classifier and Executor the invalid reason.

In practice to ensure fast response, we restrict the loop iterations into 5. For latter iterations, a lower score would be seen as valid: we use a simple linear scheduling of the passing score with the iteration index. If all iterations fail, we ask the LLM to choose between one answer provided before. More details about failure cases and number of iterations in practice can be found in Appendix A.4.

### 4.2 Tools used

For PDF Retrieval, we follow the well-practiced method of using a sentence encoder to encode chunks from guide files into a vector database and retrieve top-K closest chunks in latent space as related information given by the files. We follow Pdf-gpt (Tripathi, 2023) and utilize Universal Sentence Encoder (Cer et al., 2018) as the embedding model. For Database manipulation, we use the wrapped APIs as available tools.

## 5 Experiments

### 5.1 Metrics

Our study evaluates the model's performance through several metrics: **Instruction Set Accuracy**, **Guiding File Question Accuracy**, and **Input/Output character consumed per Question**. More details ref to Appendix A.3

We make a comparison at the character level since different LLMs utilize different tokenization methods. Input characters and output characters are separated since generating output tokens is more resource-consuming than reading input tokens for LLMs (and are more expensive in terms of API price). For clarity, we define the first two metrics mathematically.

Both **Instruction Set Accuracy** and **Guiding File Question Accuracy** are measured as:

$$\text{Accuracy} = \frac{N_{correct}}{N_{total}} \tag{1}$$

Where:

- $N_{correct}$ is the number of questions correctly answered by the model.
- $N_{total}$ is the total number of questions and instructions.

We validate whether the answer is correct by a combination of GPT4-based evaluation and human verification afterward.

This formula encapsulates the model's efficiency in accurately processing and responding to queries based on the instructions provided or the information contained within the guiding files. The metrics are instrumental in evaluating the model's adeptness at interpreting and acting upon specific sets of instructions and its capacity to extract and utilize knowledge from guiding documents, thereby providing a multidimensional view of its performance in realistic scenarios.

## 5.2 Main Experiment Results

| Architecture | LLM | $Acc_{sys}$ | $Acc_{file}$ | rela. cost. | $\#char_{in}^{avg}(k)$ | $\#char_{out}^{avg}(k)$ |
|---|---|---|---|---|---|---|
| Executor Only | gpt-4[†] | 85.6 | 83.3 | 100% | 12.9 | 0.19 |
| 1-vote CoT | gpt-3.5-turbo | 91.3 | 62.7 | **16.9**% | 14.51 | 0.5 |
| 4-vote CoT | gpt-3.5-turbo | 90.2 | 77.4 | 61.0% | 53.54 | 0.94 |
| 16-vote CoT | gpt-3.5-turbo | 90.2 | 78.4 | 239.5% | 211.58 | 2.59 |
| C-E-V | gpt-3.5-turbo | 95.2 | 90.2 | 34.4% | 30.1 | 0.56 |
| C-E-V | Mixed[‡] | **98.0** | **99.0** | 96.6% | 16.86+9.79[*] | 0.33+0.21[*] |

Table 1: Main Experiment Result.[†]: gpt-4-0125-preview is used. [‡]: gpt-3.5-turbo is used for Classifier and Verifier, while gpt-4-0125-preview is used for Executor. [*]: gpt-3-turbo characters + gpt-4-0125-preview characters. Pricing for calculating relative cost is the price obtained from OpenAI in Mar.2024. See Appendix A.2 for more details about calculating relative cost.

We evaluate the proposed **CHOPS** agent architecture on the *CPHOS-dataset*. We perform prompt tuning on gpt-4 (gpt-4-0125-preview, labeled Executor Only) and use multivote CoT (on gpt-3.5-turbo) as our baselines. We show our main experiment results in Table 1.

Compared to the CoT method, our method achieves higher accuracy while maintaining a decent cost. Multi-vote CoT does see an improvement in performance when increasing the number of votes, but it cannot provide satisfactory performance even with more cost.

Moreover, **our proposed C-E-V agent architecture with gpt-3.5-turbo as LLM backbones for all agents surpasses the Executor Only gpt-4 baseline by a large margin (**$86\% \rightarrow 95\%, 83\% \rightarrow 90\%$**) with** 34.4% **cost.** Moreover, **by substituting the Executor backbone with gpt-4, we can reach accuracy of above** 98% **on both accuracy metrics**, hugely surpassing the plain gpt-4 baseline while even cost less than the gpt-4 baseline (see Appendix A.2 for more details about price estimation). Thus our architecture is proven to be flexible and is token-efficient, reaching a balance between cost and accuracy.

## 5.3 Ablation Studies

### 5.3.1 Effectiveness and Efficiency of our proposed classifier-executor-verifier architecture

Starting from the baseline using gpt-3.5-turbo as plain Executor, we add all designed blocks and ablate the effectiveness. Finally, we substitute the Executor backbone with gpt-4 and make a comparison to plain Executor using gpt-4 as another baseline. Detailed experiment results and figure can be seen in Table 2 and Figure 5.

Classifier is shown to both reduce token consumption and improve accuracy. A more complicated 2-Level Classifier can further improve accuracy while reducing cost. This free lunch can be explained as follows. By adding a classifier we can avoid sending long retrieved chunks into the Executor in some scenarios, thus improving RAG accuracy and reducing token consumption for LLMs (Chen et al., 2023). Note by shortening the chunks sent to classifier or by using weaker yet less expensive LLMs as Classifier we can save inference cost.

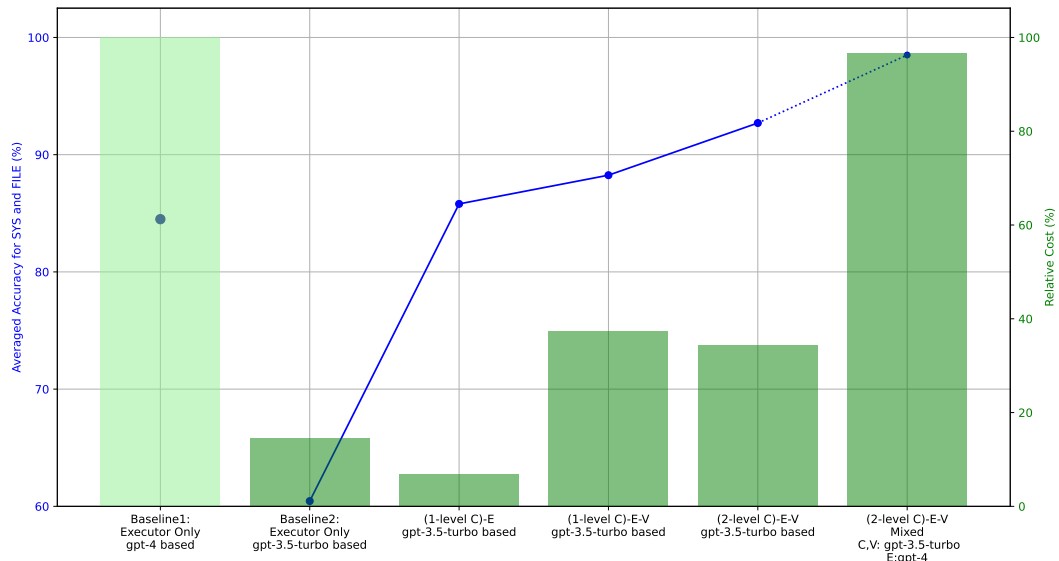

Figure 5: Effectiveness and Efficiency of 2-level Classifier, Executor and Verifier in our proposed **CHOPS-architecture**. Blue dots and lines: average accuracy for $Acc_{sys}$ and $Acc_{file}$. Green bar chart: relative cost estimated compared to Executor only with gpt-4-0125-preview backbone. Baselines: gpt-4-0125-preview and gpt-3.5-turbo.

| Architecture | LLM | $Acc_{sys}$ | $Acc_{file}$ | $\#char^{avg}_{in}$ | $\#char^{avg}_{out}$ |
|---|---|---|---|---|---|
| E | gpt-3.5-turbo | 38.5 | 82.4 | 12.8k | 0.16k |
| (1-L C*)-E | gpt-3.5-turbo | 90.4 | 81.2 | 5.98k | 0.11k |
| (1-L C*)-E-V | gpt-3.5-turbo | **96.1** | 80.4 | 32.9k | 0.51k |
| (2-L C**)-E-V | gpt-3.5-turbo | 95.2 | **90.2** | 30.1k | 0.56k |
| E | gpt-4 | 85.6 | 83.3 | 12.9k | 0.19k |
| (2-L C**)-E-V | gpt-3.5-turbo + gpt-4*** | **98.0** | **99.0** | 16.86k+ 9.79k† | 0.33k+ 0.21k† |

Table 2: Effectiveness and Efficiency of 2-level C-E-V architecture and use of mixing LLMs. APIs of Mar.2024 version are used. gpt-4-0125-preview model is used for gpt-4. *: 1-Level Classifier. **:2-Level Classifier. ***: gpt-3.5-turbo is used for Classifier and Verifier, while gpt-4-0125-preview is used for Executor. †: gpt-3.5-turbo character + gpt-4-0125-preview character.

Self-Verification is a proven effective method to improve accuracy in previous works (Weng et al., 2023). In our experiments, it has been shown to improve accuracy while consuming more tokens. However, we find that in the case where gpt-4 is used for Executor, weaker and less expensive LLMs (i.e. gpt-3.5-turbo) for Verification is enough for the architecture to produce results at very good accuracy (98%), while maintaining a decent cost at the same time.

In general, the proposed Classifier and Verifier are dealing with easier tasks but can effectively improve accuracy. Even in scenarios where very high accuracy (98%) is required and we have to use more powerful LLMs as Executor, **weaker and cheaper LLMs can still be used as Classifier and Verifier to reduce total cost while achieving satisfactory accuracy.**

## 6 Conclusion

Targeting the important scenario of Customer Service, we have collected, processed related data and proposed our *CPHOS-dataset*. Furthermore, we proposed **CHOPS-architecture**, a Classifier-Executor-Verifier agent architecture that **CH**at with cust**O**mer **P**rofile in **S**ystems, offering a flexible architecture for Customer Service scenarios. Our experiments have shown that this architecture (1) improves accuracy while controlling token consumption, achieving better accuracy compared to naively using state-of-the-art LLMs, and (2) provides a flexible architecture to utilize different LLMs for agent tasks with different levels of requirements, thus achieving satisfying accuracy with decent cost.

However, though this architecture is flexible and is not domain-specific to Customer Service in Olympiad domain and we expect it not hard to apply it to other Customer Service data, more datasets with QA pairs and Database for Customer Service are needed to further evaluate the effectiveness of our **CHOPS-architecture**. We hope future works may further augment our *CPHOS-dataset* based on the guide files, database and APIs we provide, or propose larger real-world datasets targeting the scenario of Customer Service. We are also looking forward to future work that further improves **efficiency and effectiveness** in the application of Customer Service with LLMs.

## Acknowledgement

The authors acknowledge the help with the data source from CPHOS (https://cphos.cn), an **academic non-profit organization** dedicated to providing Physics Olympiad Simulations for high school students for free. Founded in late 2020 by around 10 Physics Olympiad contestants, it now has over 100 members with 1000+ students from 100+ high schools participating in most Olympiads held. The authors also acknowledge members from CPHOS with whom they have had meaningful discussions, including: Xiaoyu Xiong, Xiangchen Tian, Zicheng Huang, Hongyi Liu, Hanyi Li, etc.

In its development process, the project acted as a course project for the NLP course advised by Professor Zhilin Yang and the TAs at Tsinghua University. The authors also acknowledge the valuable advice they provided to the project.

The authors also thank anonymous reviewers, ACs and PCs at COLM who have provided valuable suggestions that helped to improve this work.

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

## A  Appendix

### A.1  A Real-Scene dataset: CPHOS: Cyber Physics Olympiad Simulations

In the realm of customer service, datasets such as the Customer Support on Twitter dataset (Vector, 2018), which gathers over 3 million tweets and replies from prominent brands on Twitter, and the Recommender Systems and Personalization Datasets McAuley, which compile a variety of user/item interactions, ratings, and timestamps, have been instrumental. The core task within customer service can be articulated as providing accurate responses or executing specific commands in response to a user's queries or directives. This involves leveraging guiding documents or appropriately utilizing APIs. Prior datasets in file-based question answering have concentrated on reading comprehension tasks, as seen in Rajpurkar et al. (2016) and Joshi et al. (2017). Meanwhile, datasets focusing on API calls, such as Patil et al. (2023); Qin et al. (2023); Tang et al. (2023), have primarily emphasized the use of extensive API collections for task completion, intricate reasoning, and solving mathematical problems.

However, existing datasets in customer service predominantly lack detailed information about internal guiding documents or systems that can be interacted with. Similarly, datasets dedicated to file QA or API calling seldom address the customer service domain specifically.

To address this gap, we introduce the *CPHOS-dataset*, derived from the real-world context of the online platform *CPHOS: Cyber Physics Olympiad Simulations*. This dataset serves as the evaluation ground for our newly proposed **CHOPS-architecture**, emphasizing its practical application in the *CPHOS-dataset*.

*CPHOS: Cyber Physics Olympiad Simulations* is a non-profit organization focusing on physics education, predominantly operated by approximately 100 college student volunteers, primarily from China. It organizes simulated high school Physics Competitions, akin to the International Physics Olympiad (IPhO), attracting around 1000 participants from hundreds of schools annually. The organization utilizes an online system and a mini-program-based

frontend for operational tasks such as uploading and marking answer sheets. The system's backend is supported by a MySQL database, maintaining records on team leaders, vice team leaders, contestants, and examination details. Furthermore, *CPHOS* offers PDF guides on utilizing the mini-program for administrative purposes. The communication between team leaders and *CPHOS* liaison members typically relies on this documented information. After thorough data desensitization, the database, guiding documents, and QA pairs offer a representative and functional scenario for integrating LLMs into customer service, utilizing existing systems and guiding documents.

The *CPHOS-dataset* comprises a database, several guide files in PDF format, and QA pairs. These components collectively facilitate understanding how team leaders and vice team leaders can navigate the mini-program for various tasks, including uploading and grading answer sheets and accessing student grades. QA pairs are derived from real-world interactions and are further augmented by both human efforts and LLMs (including GPT-3.5 (OpenAI, 2022) and GPT-4 (OpenAI, 2023)), which enriches the dataset, making it a robust resource for validation and experimentation. The dataset's inception, modification, translation, and augmentation have been meticulously undertaken by us, with raw data sourced from *CPHOS*.

It is worth noting that while the dataset is domain-specific, the nature of the queries and instructions it encompasses—ranging from inquiries about the contents of PDF guides to requests for system profile modifications—are emblematic of the broader customer service sector. This suggests that methodologies proven effective within this dataset have the potential for broader applicability across various customer service domains, contingent upon the adaptation of guiding documents and APIs to suit specific industry requirements.

**Database.** There are 9 tables in our database, listed as following Table 3:

| table name | Description | Fields |
|---|---|---|
| cmf_tp_member | all users | id, p_id, user_name, school_id, subject, status, type, limit, create_time, nickname |
| cmf_tp_admin | all admin users | id, user_id |
| cmf_tp_school | all schools | id, area, school_name |
| cmf_tp_area | all areas of schools | id, area |
| cmf_tp_correct | all answer sheets | id, user_id, p_id, grade, status, create_time |
| cmf_tp_exam | all exams | id, status, title, type, show, create_time |
| cmf_tp_subject | all answer subjects | id, p_id, subject, image, grade, status, create_time |
| cmf_tp_test_paper | all test papers | id, p_id, user_id, student_id, score, eight, two, create_time |
| cmf_tp_student | all students | id, user_id, name, school, grade, prize |

Table 3: Tables in the Database

There are 23 APIs. We list a few here in the following table 4. Please refer to our provided codes for further information.

**User Questions and Instructions.** The questions are collected from *CPHOS* liaison members, who are responsible for responding to (visit) team leaders. Most questions are about the use of the mini-program, and others are about requests to modify the marked problems or add/modify their status in system.

| Api | Args | Description |
|---|---|---|
| AddNewSchoolByName | userId:int, Name:str, AreaName:str | add a new school given its name and area by admin |
| MakeAllTypesToBeArbiter | ChangedUserId:int | change a specific user into Arbiter |
| GetTeacherInfoBySchoolName | userId:int SchoolName:str | get user info by school name only admin user can call this successfully |
| ChangeAllTypesUploadLimit | userId:int, Limit:int | change a user's limit for its upload limit |

Table 4: Tables in the Database

## A.2 Pricing Estimation

Price information is obtained from OpenAI in March 2024. The prices are listed in the following Table 5.

| model name | Input Price per $1M$ Token | Output Price per $1M$ Token |
|---|---|---|
| gpt-3.5-turbo | $1.5 | $2 |
| gpt-4-0125-preview | $10.0 | $30.0 |

Table 5: Pricing from OpenAI

Since token number is approximately proportional to the character number, we approximate the tokens by $token\_num \approx k * char\_num$. Therefore, the final cost would be $cost \approx k * (input\_char\_num * price\_per\_input\_token + output\_char\_num * price\_per\_output\_token)$. Since the gpt families are believed to use the same tokenization method, we use the same $k$ for gpt-3.5-turbo and gpt-4-0125-preview to calculate their cost.

## A.3 Metrics

We make a comparison at the character level since different LLMs utilize different tokenization methods. Input characters and output characters are separated since generating output tokens is more resource-consuming than reading input tokens for LLMs (and are more expensive in terms of API price). For clarity, we define the first two metrics mathematically.

Both **Instruction Set Accuracy** and **Guiding File Question Accuracy** are measured as:

$$\text{Accuracy} = \frac{N_{correct}}{N_{total}} \qquad (2)$$

Where:

- $N_{correct}$ is the number of questions correctly answered by the model.
- $N_{total}$ is the total number of questions and instructions.

We validate whether the answer is correct by a combination of GPT4-based evaluation and human verification afterward.

This formula encapsulates the model's efficiency in accurately processing and responding to queries based on the instructions provided or the information contained within the guiding files. The metrics are instrumental in evaluating the model's adeptness at interpreting and acting upon specific sets of instructions and its capacity to extract and utilize knowledge

from guiding documents, thereby providing a multidimensional view of its performance in realistic scenarios.

## A.4 Failure and Iteration Number

In experiments we repeat the iteration for at most 5 times before giving all previous answers to a summarizer for final answer. In practice, the first iteration will give the correct answer (and also terminates) in more than half cases, two iterations significantly boost accuracy for system instructions, and three iterations are enough for file-based QAs in most cases. At Table 6 we record the number of correct (positive) and wrong (negative) answers per iteration and the transition between iterations.

| iteration index | 1st | | 2nd | | 3rd | | 4th | | 5th |
|---|---|---|---|---|---|---|---|---|---|
| end(TP) | 151 | | 23 | | 9 | | 2 | | 4 |
| **#pos in ith step**/#pos→pos | **168** | 13 | **30** | 6 | **14** | 5 | **6** | 4 | **4** |
| #pos→neg | / | 4 | / | 1 | / | 0 | / | 0 | / |
| #neg→pos | / | 17 | / | 8 | / | 1 | / | 0 | / |
| **#neg in ith step**/#neg→neg | **38** | 18 | **22** | 10 | **11** | 7 | **7** | 7 | **7** |
| end(FP) | 3 | | 4 | | 3 | | 0 | | 7 |

Table 6: Number of Pos/Neg in each iteration and transition between iterations.

## A.5 More LLMs

We further do experiments on the robustness of our proposed architecture on different LLMs (GLM-3, llama-2-70b). We find it hard to restrict GLM or LLaMa-based verifier to output results in the regulated format, hence we set the Verifier and Classifier into gpt-3.5-turbo and mainly focus on experimenting with the effectiveness of substituting the LLM backbone for the Executor. Please refer to Table 7 for the results.

As shown by the experiment, GLM-3 can provide rather decent performance in this case while llama-2-70b-chat shows some difficulty generating answers in a wanted format.

| Executor Backbone | $Acc_{sys}$ | $Acc_{file}$ |
|---|---|---|
| gpt-3.5-turbo | 95.2 | 90.2 |
| GLM-3-Turbo | 93.3 | 83.3 |
| llama-2-70b-chat[†] | 87.5 | 58.8 |
| gpt-4[‡] | 98.0 | 99.0 |

Table 7: More LLMs: Classifier and Verifier are gpt-3.5-turbo. For all APIs, the version of Mar.28, 2024 APIs are used. [†]: The API version of llama-2-70b-chat is used. [‡]: gpt-4-0125-preview is used.

A specific example is LLaMA-2. During our experiments we find it hard to constrain LLaMA-2 to output in certain format: as a classifier, it tends to output multiple choices rather than one choice. As a supplement, we use the most frequent character in LLaMa output as choice and adopt LLaMa as Classifier for further experiments in Table 8.

| Classifier Backbone | $Acc_{sys}$ | $Acc_{file}$ |
|---|---|---|
| LLaMa-2-7b-chat | 7.7 | 78.4 |
| LLaMa-2-70b-chat | 97.1 | 80.4 |
| gpt-3.5-turbo[†] | 95.2 | 90.2 |

Table 8: LLaMa-2 as Classifier. The 70b-chat model gives relatively decent performance, but the 7b-chat model fails to correct itself given previous failure examples, resulting in a low accuracy for system instructions.

