# OpenReview forum: "CHOPS: CHat with custOmer Profile Systems for Customer Service with LLMs"
_colmweb.org/COLM/2024/Conference — COLM_

### Official Review · Reviewer_T4SG · 2024-05-07

**Rating:** 6
**Confidence:** 4
**Ethics Flag:** 1

**Summary:**

The authors propose a framework for Retrieval-Augmented Generation based on LLMs in the domain of customer support. The framework utilises a series of smaller and larger LLMs to retrieve information from a database and to serve this information to the customer. The database is in the form of pdf guides and a MySQL database with data on users, schools, exams and answers. The developed framework is evaluated on the customer service of a Physics olympiad for teachers and students. A series of models and frameworks is evaluated, and the dataset used for evaluation is published.

**Reasons To Accept:**

1. A realistic and useful scenario is evaluated. In the examined scenario, a customer service agent is facilitating information retrieval between a customer and a database.

2. The comparison of the different frameworks is highly informative, especially the compartmentalization of the different modules, such as the use of the Classifier, which greatly increases performance.

**Reasons To Reject:**

1. Section 5.1 (Metrics) should be expanded. The evaluation methodology needs to be clearly set before the experiments and not in the Appendix. Both the metrics and the motivation behind them needs to be expanded upon in the main paper.

2. Open-source baselines are missing. As mentioned by the authors, the Classifier and Verifier deal with easier tasks. It would be interesting to examine open-source LLMs for these tasks. (amended in rebuttal)

3. Security and inference time are not discussed. The proposed framework does not currently deal with malicious user behaviour, which needs to be a greater focus. Further, for the proposed framework to be adopted, a study and comparison on inference cost is pivotal. (amended in rebuttal)

4. The paper needs some further editing, there are multiple visual miscues present, such as:

- Some citations are missing and some have spacing issues (no space between proceeding text)
- Grammatical typos, such as: "but high accuracy are needed"
- Figures 1, 2 and 4 are too small to read

---

> ### Author Rebuttal · Authors · 2024-05-27
>
> Dear Reviewer T4SG:
>
> Thank you for a detailed and constructive review! We truly appreciate your advice! Here are our responses:
>
> **Metric Section**: We will expand the experiment setting section in main paper to clearly differentiate the two cases regarding the database and the files.
>
> **Open-Source baselines**: Some Open-Source LLMs faced the difficulty to output in constrained format (e.g. LLaMa 2-70b will output multiple choices rather than one for Classifier task), making it hard to use them as Classifier or Verifier. As a supplement, we use the most frequent character in LLaMa output as choice and adopt LLaMa as our Classifier for the following experiment. The results will be included in our paper. (E,V: gpt-3.5-turbo).
>
> **Table: Performance of LLaMa and gpt-3.5-turbo as Classifiers**
>
> |Classifier|$Acc_{sys}$|$Acc_{file}$|
> |-|-|-|
> |LLaMa-2-7b|7.7|78.4|
> |LLaMa-2-70b| 97.1|80.4|
> |gpt-3.5-turbo|95.2|90.2|
>
> Llama2-70b model gives relatively decent performance, while llama2-7b model sometimes fails to correct itself given previous failure examples, resulting in a low accuracy for system instructions.
>
> **Security, inference time and cost**: We recognized that security is important, especially regarding malicious user behaviors. As mentioned in Section 3.1, we restrict the amount of APIs and their functionality. Moreover, particular APIs are parsed with an argument representing user identity not affected by LLMs, preventing attacks like admin impersonation.
>
> We agree that inference time is crucial, but it varies due to network issues and we believe character count is a metric to measure it indirectly (which is included in Table 1, Table 2). As a supplement, we further measure average time per query with CoT-SC as baselines, for your reference: (LLM: gpt-3.5-turbo).
>
> **Table: Comparison with Baselines with Inference Time**
> | |$Acc_{sys}$|$Acc_{file}$ |#$char_{in}^{avg}$|#$char_{out}^{avg}$|$Time_{avg}$ |
> |-|-|-|-|-|-|
> |1-vote|91.2|62.7|14.51k|0.50k|5.40s|
> |4-vote|90.2|77.4|53.54k|0.94k|11.48s|
> |CEV(ours)|95.2|90.2|30.1k|0.56k|7.94s|
>
> For inference cost, we estimated relative cost with character counting (please refer to Appendix A.2) and included it in Figure 5 in the main paper. We recognize its importance, hence we will add more about our estimation method in the main paper.
>
> **Miscues presents, typos and figures**: We have polished the paper and redrawn the figures. Our paper will be more accurate and readable.
>
> Thank you again for the detailed review!

---

> > ### Comment · Reviewer_T4SG · 2024-06-06
> >
> > Thank you for the rebuttal! I have amended my score from a 4 to a 6. I particularly appreciate the note on open-source baselines, the finding on the shortcomings of LLama2 is important ("LLaMa 2-70b will output multiple choices rather than one for Classifier task").

---

> ### Comment · Area_Chair_URur · 2024-06-06
> **Reviewer, please respond to rebuttal**
>
> Reviewer, please respond to rebuttal even if your score hasn't changed. The discussion period ends Thursday

---

### Official Review · Reviewer_d31m · 2024-05-09

**Rating:** 7
**Confidence:** 4
**Ethics Flag:** 1

**Summary:**

This paper discussed the growing use of Large Language Models (LLMs) such as GPTs and LLaMas in businesses and software platforms for customer service. The authors identify a gap in the integration of LLMs with customer profiles and operational capabilities in customer service models. To address this, they propose an LLM agent called CHOPS that efficiently uses existing databases or systems to access user information and provide accurate responses while avoiding harmful operations.

The authors introduce a practical dataset, the CPHOS-dataset, collected from an online platform that organizes simulated Physics Olympiads for high school teachers and students. They propose a three-agent architecture for customer service tasks: Classifier, Executor, and Verifier (C-E-V). The Classifier classifies user requests, the Executor provides responses or executes operations, and the Verifier assesses the validity of commands executed by the Executor.

The paper also includes experiments to validate the performance of the proposed CHOPS architecture using the CPHOS-dataset. The results demonstrate that the CHOPS architecture can enhance or serve as an alternative to human customer service. The authors conclude by emphasizing the need for more datasets with QA pairs and Database for Customer Service to further evaluate the effectiveness of the CHOPS-architecture.

**Reasons To Accept:**

The paper addresses a significant gap in the integration of LLMs with customer profiles and operational capabilities in customer service models. This is a relevant and timely topic given the increasing use of AI in customer service. The introduction of the CHOPS architecture and the C-E-V model is innovative and could potentially change the way customer service is handled in businesses. The authors have done a commendable job of validating their model using the CPHOS-dataset. The results demonstrate the effectiveness of the proposed architecture.

**Reasons To Reject:**

1. The paper could benefit from a more detailed explanation of the CHOPS architecture and the C-E-V model. The current description may not be sufficient for readers unfamiliar with these concepts.
2. The authors have used the CPHOS-dataset for validation. However, the effectiveness of the model in real-world scenarios remains untested.
3. The authors could have included a comparison with other existing models. This would help readers understand how the proposed model stands against the current state-of-the-art.

---

> ### Author Rebuttal · Authors · 2024-05-27
>
> Dear Reviewer d31m,
>
> Thanks for the detailed and advisable review! We truly appreciate your advices!
>
> Here are our corresponding replies:
>
> **Description about our model**: We recognize that the current explanation is be a bit less detailed and we will add more descriptions in our next version of paper. We will add more overview and specific details, especially for the Executor and Verifier agent.
>
> **Real-world scenarios**: Our dataset is collected in a real-world scenario, and we think our model would be a real-world solution. Moreover, we wish our work may encourage more datasets with Database, APIs and related Guiding Files to be proposed in future works, and we wish to test our models to see its performance on a broader range of Customer Service. We hope our work may bring insights for LLM applications in Customer Service, not only for model structure but also for more possible real-world datasets.
>
> **Comparison with other existing models**: Recent works on agent or LLM reasoning (e.g. ToT, Cumulative Reasoning, etc) underlines efficient searching over a large state space for possible answer (e.g. math problems, etc) but Customer Service requires more about accuracy and less about reasoning skills compared to these complicated tasks, and these more complex models are hard to accommodate to this scenario. In comparison, our work addresses a key gap in integrating LLMs with customer profiles and operational capabilities in customer service models. As a supplement, we compare with CoT-SC method that is easier to adopt to the Customer Service scenario, and compare the CHOPS C-E-V architecture with it, as follows:
>
> **Table: Comparison with CoT-SC**
>
> |  | $Acc_{sys}$ | $Acc_{file}$ | $Acc_{avg}$ | #$char_{in}^{avg}$ | #$char_{out}^{avg}$ |
> | ---- | --- | ----- | ---- | -------- | ------- |
> | 1-vote CoT   | 91.2        | 62.7  | 77.0 | 14.51k   | 0.50k   |
> | 4-vote  CoT-SC  | 90.2| 77.4    | 83.8        | 53.54k | 0.94k     |
> | 16-vote CoT-SC  | 90.2  | 78.4   | 84.3        | 211.58k  | 2.59k     |
> | CEV (ours) | 95.2   | 90.2    | 92.7      | 30.1k      | 0.56k      |
>
> (We will include this table in our paper. All models use gpt-3.5-turbo)
>
> Compared to CoT-SC, our proposed architecture achieves higher accuracy with less cost.
>
> Thank you for the detailed review!

---

> > ### Comment · Reviewer_d31m · 2024-06-05
> >
> > Thanks for providing more details.

---

### Official Review · Reviewer_Utie · 2024-05-11

**Rating:** 7
**Confidence:** 4
**Ethics Flag:** 1

**Summary:**

The paper proposes a framework called CHOPS (CHat with custOmer Profile in existing System) that aims to integrate large language models (LLMs) into customer service systems that involve guiding files and databases. The key components are: 1) A new dataset called CHOPS-dataset derived from a real-world online platform for physics Olympiads, including a database, PDF guides, and user questions/instructions. 2) A classifier-executor-verifier architecture where the classifier determines what information (database, guides) is needed, the executor generates a proposed answer/operation using that information, and the verifier checks if the executor's output is valid. 3) Using different LLMs for each component to balance accuracy and cost - e.g. a powerful LLM like GPT-4 for the executor but cheaper models like GPT-3.5 for the classifier/verifier. The authors conduct extensive experiments showing their architecture can achieve high accuracy (98%) at reasonable cost on the CHOPS-dataset by combining LLMs in this way.

**Questions To Authors:**

1. The paper mentions iterating the pipeline up to 5 times if the verifier rejects outputs. How often do multiple iterations end up being required in practice?
2. The results focus on English inputs/outputs. How might you handle customer service scenarios requiring other languages? Would the architecture need to be modified?

**Reasons To Accept:**

1. The paper proposes a novel application of integrating LLMs into real-world customer service systems with databases and guidance in a accurate and cost-effective manner.
2. New dataset (CHOPS-dataset) derived from practical scenarios to facilitate research in this area.
3. Thorough experiments evaluating the different components and comparing performance/cost trade-offs.
4. The classifier-executor-verifier architecture is a promising approach that could generalize beyond just customer service.

**Reasons To Reject:**

1. Results are solely on the CHOPS dataset, which is quite specific and limited. More experiments on other customer service domains would strengthen the claims.
2. Limited analysis of why the classifier-executor-verifier works well and what kinds of errors/failures occur.
3. The database/API setup is complex and not easily reproducible from the paper.

---

> ### Author Rebuttal · Authors · 2024-05-27
>
> Dear Reviewer Utie,
>
> Thank you for a detailed and inspiring review! We truly appreciate your advices and questions! Here are our responses:
>
> **Dataset limitation**: Our results are based on the specific dataset we proposed.
> We hope our work may encourage more practical dataset with database, files and QA pairs for Customer Service, and we are looking forward to evaluating CHOPS architecture on future datasets.
>
> **Failure cases analysis**: A false positive in the Verifier stage leads to an incorrect output, while errors in the Classifier or Executor stages may be corrected by the Verifier in subsequent iterations. Classifier errors are the most frequent but are easy to identify and correct with the Verifier at the 2nd iteration. For difficult questions, the Executor may fail, and these failures are harder to correct and need more iterations: please refer to the following table about iteration number. We will add more analysis in the next version of our paper.
>
> **Complex setup procedure**: We recognize the database we include in our proposed dataset makes the setup with some workload, but using database like SQLAlchemy that can be easily installed will improve reproducibility, which will be a good choice for the next version of our code.
>
> **Iteration number used in practice**: More than half of the queries end at the first iteration, with few false positives. Two iterations significantly boost accuracy for system instructions, and three iterations do so for file questions. We use five iterations to ensure completion. Here is a table for reference:
>
> **Table: Pos/Neg in each Iteration and Transition between Iterations**
>
> ||1st||2nd||3rd||4th||5th|
> |:----:|:-:|:--:|:---:|:--:|:---:|:-:|:--:|:-:|:--:|
> |end(TP)|151||23||9||2||4|
> |**\#pos in ith step**/\#pos->pos|**168**|13|**30**|6|**14**|5|**6**|4|**4**|
> |\#pos->neg|/|4|/|1|/|0|/|0|/|
> |\#neg->pos|/|17|/|8|/|1|/|0|/|
> |**\#neg in ith step**/\#neg->neg|**38**|18|**22**|10|**11**|7|**7**|7|**7**|
> |end(FP)|3||4||3||0||7|
>
>
> **Support for multiple languages**: Our architecture will only require slight modifications as long as Neural Networks with multilingual support are used. LLMs supporting targeted languages and a multilingual sentence encoder for file RAG are needed. Further edits may include prompt tuning for the Verifier to respond in the corresponding language.
>
> Thank you again for your detailed review!

---

> > ### Comment · Reviewer_Utie · 2024-06-06
> >
> > Thanks for the response. It's helpful.

---

> ### Comment · Area_Chair_URur · 2024-06-06
> **Reviewer, please respond to rebuttal**
>
> Reviewer, please respond to rebuttal even if your score hasn't changed. The discussion period ends Thursday

---

### Author Response · Authors · 2024-06-06
**Thanks for reviewer responses; Open for further discussions**

Dear Reviewers,

Thank you for taking time to do reviews and responses! We hope our previous feedback provide more supplementary information about our work and address your concerns.

We are also looking forward to possible further discussions in the remaining discussion period!

---

### Author Response · Authors · 2024-06-06
**Summary of Revisions**

We sincerely thank all reviewers for their detailed comments, which are very constructive for our work.

In our work, we proposed a real-world Customer Service dataset with database, guiding files and QA pairs. We also proposed the CEV agent-based architecture for utilizing LLMs on Customer service and evaluated it on our proposed dataset. We hope our work may bring insight for LLM applications in Customer Service in the aspect of agent architecture and encourage the proposal of more real-world datasets with databases, APIs, related guiding files and other related materials as well.

The reviewers generally hold positive opinions to our paper, in that our work proposes a novel and timely application of LLMs in the real-world scenario of Customer Service, our experiments are informative that show the effectiveness of our proposed architecture on performance/cost trade-offs, and our proposed architecture provide insight that may generalize beyond Customer Service.

The reviewers have also raised detailed and constructive concerns that help us improve our work. Compared to the original version, our new version of paper will feature enhancements in the following aspects:

**Supplementary experiment results and analysis**

1. The table for Comparison with baselines will be added to the main paper. (Reviewer d31m, Reviewer T4SG)
2. The table for Performance of open-source llms as Classifiers as well as analysis about shotcomings of Llama2 will be added. (Reviewer T4SG)
3. More analysis about failure case and iteration number required in practice will be added. (Reviewer Utie)

**More detailed explanations in the main paper**

4. More detailed explanation about the CEV model will be added in the main paper. (Reviewer d31m)
5. More details about experiment settings will be added in the main paper. (Reviewer T4SG)
6. More details about estimation methods for inference cost will be added in the main paper. (Reviewer T4SG)

**Amendment for Miscues presents**

7. Miscues presents, typos will be corrected and figures will be redrawn for readability. (Reviewer T4SG)

We will also modify current codebase to simplify setup procedure for better reproducibility. (Reviewer Utie)

The valuable suggestions from reviewers are very helpful for us. We would be glad to have further discussions.

---

### Decision · Program_Chairs · 2024-07-10

**Decision:**

Accept

**Comment:**

This paper introduces a system for using a Large Language Models (LLMs) in the application of customer service. Due to the number of customer support agents in the word, this could have a significant effect. The new framework, CHOPS, combines small and large LLMs to work with existing customer profiles, making customer service more efficient. They tested this with a new dataset from a real-world online platform for Physics Olympiads.

Pros:
* Novel Idea: It addresses a gap in current LLM-based customer service by integrating them better with customer profiles.
* New Dataset: The CPHOS-dataset is a valuable resource for future research.
* Thorough experiments evaluating the different components and comparing performance/cost trade-offs.

Cons:
* Further testing on other customer service domains would help show it works more broadly.
* Additional details on the CHOPS architecture would help readers understand it better.
* Additional editing is required for the paper.

In summary, all reviewers liked the paper and thought it should be accepted.